# Quantifying and comparing radiation damage in the Protein Data Bank

Kathryn L. Shelley [1,2✉] & Elspeth F. Garman [1✉]

Radiation damage remains one of the major bottlenecks to accurate structure solution in protein crystallography. It can induce structural and chemical changes in protein crystals, and is hence an important consideration when assessing the quality and biological veracity of crystal structures in repositories like the Protein Data Bank (PDB). However, detection of radiation damage artefacts has traditionally proved very challenging. To address this, here we introduce the $B_{net}$ metric. $B_{net}$ summarises in a single value the extent of damage suffered by a crystal structure by comparing the $B$-factor values of damage-prone and non-damage-prone atoms in a similar local environment. After validating that $B_{net}$ successfully detects damage in 23 different crystal structures previously characterised as damaged, we calculate $B_{net}$ values for 93,978 PDB crystal structures. Our metric highlights a range of damage features, many of which would remain unidentified by the other summary statistics typically calculated for PDB structures.

[1] Department of Biochemistry, University of Oxford, South Parks Road, Oxford OX1 3QU, United Kingdom. [2] School of Chemistry, University of Bristol, Cantock's Close, Bristol BS8 1TS, United Kingdom. ✉email: kathryn.l.shelley@gmail.com; elspeth.garman@bioch.ox.ac.uk

The radiation damage suffered by crystals during X-ray diffraction experiments has long hindered accurate structure determination in protein crystallography. Despite the development of various damage mitigation strategies, including cryo-cooling[1] and optimisation of the data collection strategy using software such as BEST[2] or RADDOSE-3D[3], data collection methodologies using the increasing flux densities of synchrotron light sources have resulted in radiation damage remaining one of the major challenges in protein crystallography. This problem will only be further exacerbated as the recently constructed fourth-generation synchrotron sources come online.

Radiation damage can affect both the individual asymmetric unit copies and the overall crystal lattice structure. Damage to the crystal lattice (global radiation damage) is readily detectable from changes in diffraction pattern reflection intensities, primarily as the fading and ultimate loss of high-resolution reflections[4]. Crystallographers are therefore able to truncate their datasets to discard diffraction patterns substantially affected by global radiation damage (within the limitation of retaining sufficient data completeness for structure solution).

Conversely, damage to the individual asymmetric unit copies (specific radiation damage) has traditionally proven challenging to detect within individual protein crystal (PX) structures. Accordingly, specific radiation damage is usually studied by identifying differences between successive datasets collected from the same crystal(s) (e.g. the radiation damage datasets collected from six different proteins and deposited by Nanao et al.[5]). Differences are caused by structural rearrangements (chiefly side-chain disordering) and chemical changes initiated by the electrons that are ejected upon the absorption of incident X-rays by the sample. At cryotemperatures (around 100 K), these induced chemical changes have been observed to occur in a reproducible order with increasing dose (the energy absorbed per unit mass of the crystal): for example, in PX structures metal ions are the first to be reduced[6]; followed by the breakage of disulfide bonds; aspartate and glutamate residues are then decarboxylated; and next the methylthio group is cleaved from methionine residues[7–9].

Consequently, in contrast to global radiation damage, crystallographers are often unable to detect specific radiation damage artefacts within their structures and correct them appropriately. As an example, one could assume that an active site glutamate residue is disordered and that such disordering is potentially involved in the catalytic mechanism of its parent enzyme, when in fact the residue has just been decarboxylated by the incident X-rays. Such errors can compromise the conclusions drawn from a structure, and necessitate studies to separate biologically relevant features from those caused by radiation damage (e.g. during the bacteriorhodopsin photocycle[10,11], and the recent study of the bending of flavin in the mechanism of fatty acid photodecarboxylase[12]).

Furthermore, and unfortunately, specific damage usually onsets prior to global damage: the experimental dose limit (corresponding to a 30% loss in summed reflection intensities from apo- and holo-ferritin crystals) was reported as 30 MGy at 100 K[13], whereas aspartate/glutamate decarboxylation has been detected at doses as low as 4 MGy[14]. Owing to the difficulties associated with its detection, the number of Protein Data Bank (PDB)[15] structures containing specific radiation damage artefacts is unknown; however, given that during a typical X-ray diffraction experiment a protein crystal held at 100 K absorbs a dose on the order of 1–10 MGy per complete dataset[4], it is likely to be a substantial fraction.

In light of these issues, previously our group developed a metric called $B_{Damage}$, a per-atom quantity that identifies potential sites of specific radiation damage within individual atomic resolution PX models as atoms with high full isotropic atomic $B$-factor (from herein referred to as $B$-factor) values relative to other atoms in a similar packing density environment[16]. $B_{Damage}$ has been validated to highlight known sites of specific damage in structures derived from a range of different radiation damage datasets. It is therefore a useful tool to flag up suspect sites within individual PX structures.

Unfortunately, however, the variability in the relationship between $B$-factors and radiation damage, caused by factors such as differences in refinement protocols and data resolution, means that absolute $B_{Damage}$ values cannot be fairly compared between structures. Furthermore, it is not possible to determine a single threshold value of $B_{Damage}$ beyond which an atom has a fixed probability of being damaged. These shortcomings prevent $B_{Damage}$ from being used to systematically assess the radiation damage suffered by the PX structures deposited in the PDB. Consequently, specific radiation damage is currently largely ignored in PX structure quality assessment.

To address these problems, in this work we have developed the $B_{net}$ metric. $B_{net}$ is a derivative of $B_{Damage}$ that summarises the overall extent of damage suffered by a PX structure in a single value. Initially, we demonstrate that $B_{net}$ values can be fairly compared between different PX structures, before using the metric to quantify for the first time the extent of radiation damage suffered by a dataset of 93,978 PX structures in the PDB. Based upon this analysis, we propose the $B_{net}$ metric to be a highly useful tool to assess PX structure quality for both newly and previously deposited structures.

## Results

**Defining the $B_{net}$ metric.** When a structure suffers specific radiation damage, the $B$-factor values of the affected atoms increase relative to unaffected atoms (provided that the affected sites are not subjected to occupancy refinement). The $B_{Damage}$ metric identifies potentially damaged atoms as those with high $B$-factor values in comparison to other atoms in a similar local environment (Fig. 1 and Methods). Consequently, as a structure is increasingly damaged, the $B_{Damage}$ values of its affected atoms increase, and furthermore the distribution of the structure's $B_{Damage}$ values becomes increasingly positively skewed. The skewness of its $B_{Damage}$ distribution can thus be used as a measure of the overall extent of specific radiation damage suffered by an individual PX structure. However, specific radiation damage typically only affects a minority of the atoms in even the most extensively damaged structures. Accordingly, the sensitivity of a metric that measures the skewness of the $B_{Damage}$ distribution of all atoms in a structure to specific radiation damage artefacts was found to be suboptimal (Supplementary Fig. 1).

Therefore, we have instead focussed upon the $B_{Damage}$ values of known sites of specific damage. Because different damage events occur at different rates, plus affect different relative proportions of atoms in different structures, in order to obtain a value that is fairly comparable between structures we decided to focus upon only one type of damage event. We have selected the decarboxylation of aspartate and glutamate side chains because: (i) the vast majority (99.8% (3sf)—see Methods for details on how this and subsequent dataset sizes plus percentages were calculated) of PX structures deposited in the PDB contain one or more aspartate/glutamate residues; and (ii) aspartate/glutamate decarboxylation is one of the earliest onset and most extensively studied specific radiation damage events. Because aspartate/glutamate decarboxylation has primarily been characterised and studied in cryo-cooled crystals, and in fact has thus far not been convincingly observed at room temperature[17], we restrict the following validation of our $B_{net}$ metric and subsequent analysis of the PDB to structures collected at 80–120 K (88.0% (3sf) of PX structures in the PDB).

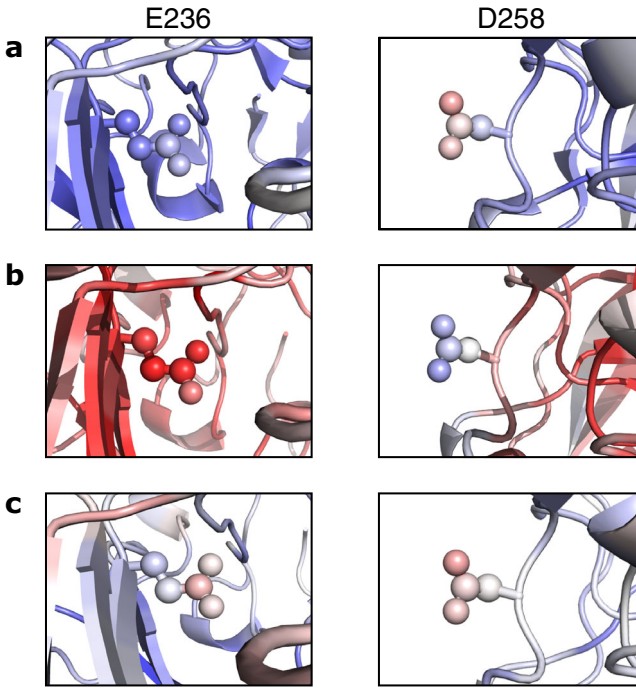

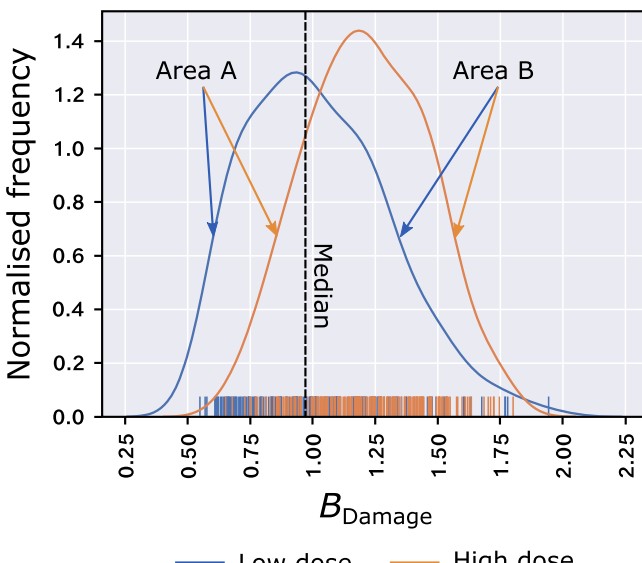

**Fig. 1 Illustration of the $B_{Damage}$ metric.** A damaged structure of GH7 family cellobiohydrolase (PDB accession code 5MCN[35]) is coloured (blue: low value; red: high value) by **a** *B*-factor, **b** packing density and **c** $B_{Damage}$. The side chains of E236 and D258 are shown (as spheres) in the left- and right-hand images, respectively; although they have different *B*-factor values because the local packing density environments of the two residues are different, their $B_{Damage}$ values reveal the two residues to be similarly damaged. An atom's packing density is defined as the number of non-hydrogen atoms within a 7 Å radius in the context of the crystalline structure. For visual clarity, only a single asymmetric unit copy is shown, however packing density and $B_{Damage}$ values have been calculated in the context of the crystalline structure as described in the Methods.

**Fig. 2 Calculation of the $B_{net}$ metric.** The $B_{net}$ metric is calculated from the kernel density estimate of the $B_{Damage}$ values of a structure's aspartate and glutamate side-chain oxygen atoms as the ratio of the area under the curve on either side of the median $B_{Damage}$ value of all atoms in the structure. Displayed kernel density estimates and rug plots are of a low (1.11 MGy, PDB accession code 5MCC) and a high (22.7 MGy, PDB accession code 5MCN) dose structure of GH7 family cellobiohydrolase[35]. A and B are the areas under the kernel density estimate of the $B_{Damage}$-Asp/Glu distribution to the left- and right-hand side of the median $B_{Damage}$ value of all atoms in the structure, respectively. Source data are provided in a Source Data File.

Accordingly, the $B_{net}$ metric measures the skewness of the $B_{Damage}$ values of a structure's aspartate and glutamate side-chain carboxyl group oxygen atoms (the $B_{Damage}$-*Asp/Glu distribution*) in relation to the $B_{Damage}$ values of all atoms in the structure. Specifically, $B_{net}$ is calculated from the kernel density estimate of the $B_{Damage}$-Asp/Glu distribution plus the median $B_{Damage}$ value of all atoms in the structure: it is defined as the ratio of the area (B) under the curve on the right-hand side of the median to the area (A) under the curve on the left-hand side of the median (Eq. 1 and Fig. 2).

$$B_{net} = \frac{B}{A} \qquad (1)$$

Experimentally, differences in resolution and refinement protocols between PX structures mean that *B*-factor values do not scale with damage via any consistently observed relationship. Consequently, although all $B_{Damage}$ distributions have a similar shape (approximately lognormal about 1), the standard deviation and range of the $B_{Damage}$ distribution is largely independent of the extent of damage suffered by a structure; hence $B_{Damage}$ values can be fairly compared within, but not between, structures.

We have therefore defined $B_{net}$ as in Eq. 1, since, in comparison with other measures of skewness such as quartile-based methods and Pearson's skewness coefficients, this area ratio is more strongly influenced by the number of $B_{Damage}$ data points on either side of the median, as opposed to their raw values (and correspondingly the standard deviation and range of the

distribution). Moreover, using such an area-based calculation makes fewer assumptions about the shape of the $B_{Damage}$-Asp/Glu distribution; unlike their all-atom counterparts, there can be substantial variation between the shapes of the $B_{Damage}$-Asp/Glu distributions of different (especially smaller) structures. We compared five different methods of measuring skewness and found that $B_{net}$ as defined in Eq. 1 achieved the best performance on our dataset of PDB structures (see Methods for further details). This analysis is available at https://doi.org/10.5281/zenodo.5567741.

In the following analysis, we validate that $B_{net}$ values can be fairly compared between different PX structures.

**Validation of the $B_{net}$ metric.** In order for $B_{net}$ to be a valid measure of specific radiation damage that can be fairly compared between different structures, it must: (i) scale, via a consistent relationship, with the radiation damage suffered by a structure; and (ii) be independent of other variables that can affect *B*-factor values.

**$B_{net}$ increases with dose.** To test whether $B_{net}$ satisfies the first of these conditions, we calculated $B_{net}$ values for 23 radiation damage datasets[18–35] deposited in the PDB as of 19 November 2020 (Supplementary Tables 1 and 3). To the best of our knowledge, we have included in our analysis every radiation damage dataset that: (i) resulted in a protein structure (owing to a lack of suitable radiation damage datasets, $B_{Damage}$ has not been validated for nucleic acids); (ii) was collected at a temperature in the range of 80–120 K; (iii) contains data for three or more structures collected at increasing doses; iv) each dataset in the series has been collected from the same crystal(s) (hence successive structures are definitely increasingly damaged); and v) has

associated information about the dose absorbed during each dataset in the series, reported in gray (Gy). Because $B_{net}$ is a $B$-factor-derived metric, we also excluded datasets resulting in structures that contain one or more aspartate/glutamate residue(s) whose occupancy across all listed conformers is less than one. For the same reason, we only retained datasets giving models which were refined with per-atom $B$-factors (either isotropic or anisotropic), and discarded those refined with e.g. per-residue $B$-factors or similar. Additionally, we excluded datasets giving one or more structures with a resolution worse than 3.5 Å; as resolution deteriorates, the number of observed reflections decreases, meaning fewer model parameters can be refined. It is highly unlikely that per-atom $B$-factor refinement is the appropriate choice for structures solved to worse than 3.5 Å resolution. Lastly, we removed datasets giving one or more structures with an $R_{free}$ value greater than 0.4, since these models show such poor fit to the electron density data that detection of radiation damage artefacts is very difficult if not impossible.

The extent of radiation damage suffered by a PX structure at cryotemperatures is proportional to the dose absorbed by the crystal[36], as initially observed at room temperature by Blake and Phillips in 1962[37]. Therefore, if indeed it is a valid measure of radiation damage, we expect $B_{net}$ to increase monotonically with dose, and furthermore to obtain the same $B_{net}$ value for different structures exposed to the same dose.

As shown in Fig. 3, we observe an approximately linear relationship between $B_{net}$ and dose for aspartate/glutamate side-chain oxygen atoms. This is not without precedent; many studies have previously observed linear correlations between a range of different $B$-factor-based metrics and dose[7,38,39]. However, unlike these other relationships, the correlation between $B_{net}$ and dose is much more consistent across the 23 datasets, since the gradients and y-intercepts of the lines of best fit plotted through the datasets are largely similar, as measured by their relative standard deviations (Fig. 3, plus Supplementary Tables 1, 2).

Unsurprisingly, given the huge diversity of structures deposited in the PDB, there are a few exceptions. Firstly, the linear relationship between $B_{net}$ and dose is noisy (with some datasets being affected more than others): $B$-factors can be affected by multiple variables in addition to radiation damage. The main such variable is mobility: however other researchers have previously demonstrated how relative $B$-factor values can also be distorted by map over-sharpening, grossly different mobilities of subunits in the same asymmetric unit, and poor fit of the model to the data, amongst other variables[40]. Although the $B_{Damage}$ metric (and likewise the derivative $B_{net}$) attempts to remove the effects of mobility, and concomitantly some of the other variables that impact upon $B$-factor values, by comparing the $B$-factor values of atoms in similar local packing density environments, this correction is not perfect. Moreover, there is a considerable error associated with dose calculations, to the extent that others have previously applied a threshold of a factor of 2 when assessing whether the variation in radiation damage effects in different experiments is significant or not[41].

Consequently, it is not surprising that the datasets resulting in fewer structures tend to have more extreme gradient and y-intercept values. In particular, the gradient of the series collected by Castellvi et al.[22] is double that of the next highest gradient. However, this dataset series only consists of three structures, hence a small change in either $B_{net}$ or dose of one or more of these structures would cause a large change in gradient.

Secondly, restraints applied during refinement prevent the $B$-factor values of neighbouring atoms from varying too much from one another. This can prevent the $B$-factor values of damaged atoms from increasing to reflect their damage. In spite of such restraints, the fact that $B_{net}$ increases with a dose for all except one

series indicates that these restraints are generally not so tight that they prevent the $B_{net}$ metric from being an effective measure of damage. However, it is notable that of the four series with a gradient of 0.05 MGy$^{-1}$ or less, two of them increase to gradients of greater than 0.1 MGy$^{-1}$ when the structures' $B$-factors are re-refined (see Supplementary Table 4 and Methods). It is unclear why the other two damage series, both from the same publication[32], show an unusually low gradient.

Lastly, we observe that the $B_{net}$ values for the series collected by Bury et al.[35] plateaus after ~14 MGy has been absorbed (consequently, we have excluded these data points when calculating the gradient and y-intercept of the line of best fit between $B_{net}$ and dose for this series). However, in contrast to their similar $B_{net}$ values, Bury et al. observe increasing damage artefacts between the final four structures in the series. Looking more closely, one can see that whilst in the earlier structures in the series the number of aspartate/glutamate side-chain oxygen atoms with $B_{Damage}$ values greater than the median (of all atoms in the structure) increases, in later structures this number remains fairly constant whereas the $B_{Damage}$ values of the damaged atoms become more extreme (Supplementary Fig. 2). A disadvantage of our method of measuring skewness is that it is less effective at picking up those latter types of changes. However, as described above, this is outweighed by the benefits of making fewer assumptions about the shape of the $B_{Damage}$ Asp/Glu distribution, especially for smaller structures: hence we have not changed our methodology. Importantly, the value of $B_{net}$ attained by the structure that has received 14.1 MGy is sufficiently high to mark it, plus the subsequent structures in the damage series, as having an unusually high $B_{net}$ value that merits further inspection, so fulfilling the purpose of the metric.

As an additional demonstration that the increasing $B_{net}$ values are representing damage to the successive structures in these radiation damage series, we have also calculated the equivalent of $B_{net}$ values for the side-chain amide group oxygen atoms of asparagine and glutamine residues ("Asn/Gln-$B_{net}$"). As can be seen in Fig. 3(b) plus Supplementary Tables 1, 2, the relationship between Asn/Gln-$B_{net}$ and the dose is much more inconsistent than for Asp/Glu-$B_{net}$. The fit of the data to a linear relationship is poor, plus the gradients and y-intercepts of the straight lines of best fit that have been plotted are highly variable, reflected by their higher relative standard deviation scores as compared to Asp/Glu-$B_{net}$ (Supplementary Tables 1, 2). Moreover, in Fig. 3(c) we plot Wilson $B$-factor vs. dose for the 23 damage series. The data largely show a good fit to a linear relationship, however, because as described earlier experimental differences prevent $B$-factors from scaling consistently with dose across different structures, there is much more variation present within the gradients and in particular the y-intercepts of the calculated lines of best fit (Supplementary Tables 1, 2). Accordingly, it is clear that the (Asp/Glu-)$B_{net}$ metric successfully detects differences between increasingly damaged structures that are not captured when measuring either Wilson $B$-factor or the $B_{Damage}$ values of residues that do not suffer chemical changes as a result of radiation damage.

Consequently, this analysis supports the conclusion that $B_{net}$ values scale with dose via a consistent relationship, independent of structure identity.

**$B_{net}$ correlates with resolution**. To test whether $B_{net}$ is independent of variables, other than dose, that can affect $B$-factor values, and hence whether $B_{net}$ solely reflects the specific damage suffered by a structure, we calculated $B_{net}$ values for the 105,856 cryotemperature (80–120 K) PX structures deposited in the PDB with accompanying experimental data as of 19 November 2020.

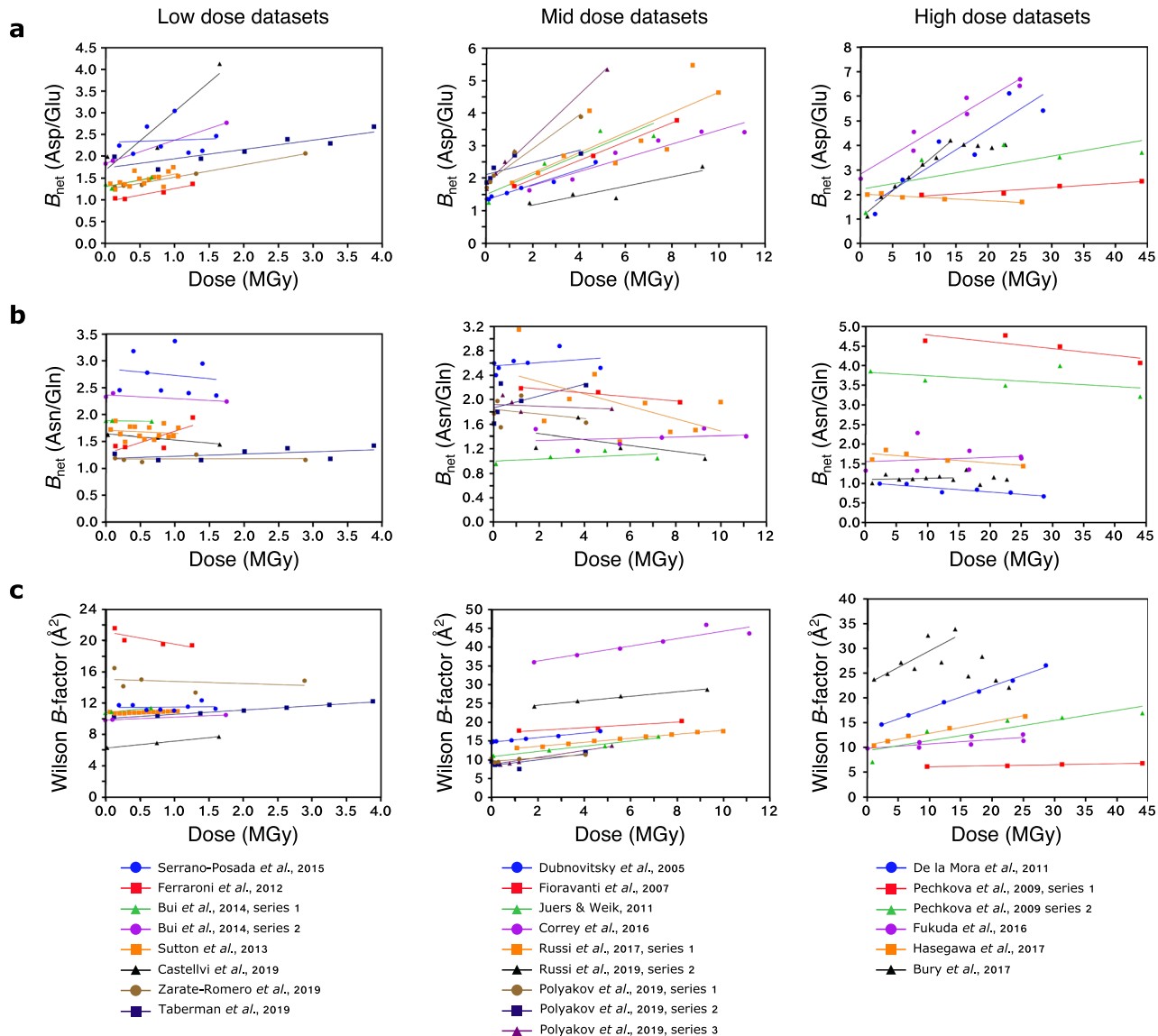

**Fig. 3 Dose dependence of the $B_{net}$ metric for radiation damage datasets. a** $B_{net}$ (Asp/Glu), **b** $B_{net}$ (Asn/Gln) and **c** Wilson $B$-factor vs. dose for 23 radiation damage datasets. Straight lines of best fit, calculated via least squares regression, are plotted through each dataset. For clarity, the datasets have been split into "low" (0–4 MGy), "mid" (4 – 12 MGy) and "high" (>12 MGy) groups, determined by the highest dose absorbed. Source data are provided at https://doi.org/10.5281/zenodo.5566557.

These structures were filtered using the same criteria listed above for the radiation damage datasets, apart from the requirements (i) to be part of a radiation damage dataset and (ii) for available dose information—the majority of structures in the PDB are deposited with neither their dose nor all of the information required to estimate that dose. In addition, we rejected structures with fewer than 20 carboxyl group oxygen atoms across their aspartate and glutamate side chains (an empirically determined threshold—see Supplementary Fig. 3), plus those which reported one or more null values for the eight variables (listed below) against which we compared $B_{net}$. We also discarded structures refined with a flat $B$-factor model (information that, unlike PDB structures, is recorded for PDB-REDO[42] structures—see below). Applying these filters left us with a dataset of 93,978 structures (i.e. 88.8% (3sf) of the initial 105,856).

For this analysis, we decided to use structures deposited in the PDB-REDO databank rather than in the PDB. PDB-REDO uses the latest available crystallographic software to re-refine all X-ray crystal structures deposited in the PDB with associated coordinates and structure factors. Consequently, structures in the PDB-REDO databank are more fairly comparable with one another, since they have been subjected to a consistent refinement protocol. Furthermore, extensive information about the final round of refinement, including the strength of the $B$-factor restraints applied, is available for all PDB-REDO structures. Note that because PDB-REDO is an automated pipeline, although in most cases it improves the structures it processes, it can also occasionally introduce errors, hence our decision to use the original PDB structures in our analysis of 23 radiation damage series in the previous section.

We calculated the (Spearman's rank) correlation coefficients between the 93,978 PDB-REDO structures' $B_{net}$ values and eight variables we predicted might impact their $B_{net}$ values in addition to radiation damage: namely, their resolution; $R_{work}$; $R_{free}$; temperature; molecular mass; the percentage and the raw number of aspartate/glutamate residues they contain; and the strength of the $B$-factor restraints applied in their final round of refinement. As discussed above, the lack of available dose information for the

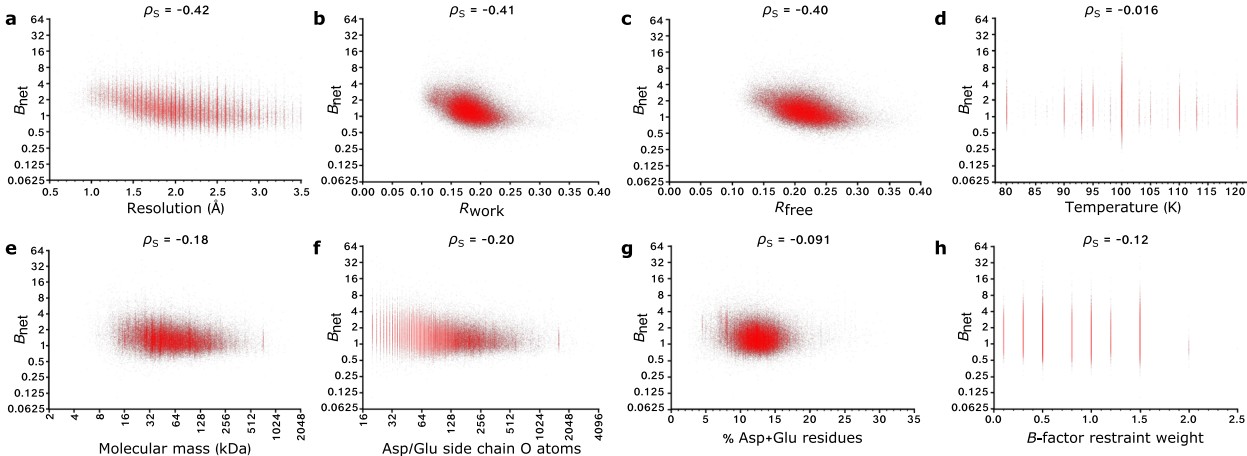

**Fig. 4 Investigating the dependence of $B_{net}$ on various parameters.** Scatter plots of $B_{net}$ vs.: **a** resolution; **b** $R_{work}$; **c** $R_{free}$; **d** temperature; **e** molecular mass; **f** the number of aspartate/glutamate side-chain oxygen atoms; **g** the percentage of aspartate/glutamate residues; and **h** the strength of $B$-factor restraints applied in the final round of refinement. The Spearman's rank correlation coefficient ($\rho_S$) between the two variables is reported for each plot. Source data are provided at https://doi.org/10.5281/zenodo.5566558.

majority of PDB(-REDO) structures has unfortunately prevented us from comparing $B_{net}$ to dose as a positive control in this analysis.

We calculated Spearman's rank correlation coefficients between each of the eight variables listed above and $B_{net}$ (Fig. 4 and Supplementary Table 5). Spearman's rank ($\rho_S$) rather than Pearson's correlation coefficient was selected since the former measures a monotonic relationship between variables, whereas the latter is a measurement of linearity. Using the threshold that absolute correlation coefficients of less than 0.2 (i.e. $-0.2 < \rho_S < 0.2$) indicate at most only a weak relationship between two variables, we observe no substantial correlation between $B_{net}$ and any of temperature, molecular mass, the number and percentage of aspartate and glutamate residues, and $B$-factor restraint weight.

However, we observe non-negligible relationships between $B_{net}$ and resolution, $R_{work}$ and $R_{free}$. The strongest correlation is between $B_{net}$ and resolution. The negative correlation between these two variables can be readily understood; as resolution decreases, electron density maps become more indistinct, and hence additional blurring due to aspartate/glutamate decarboxylation becomes more difficult to distinguish. Furthermore, a structure's $R_{work}$ and $R_{free}$ values are strongly correlated with its resolution (the $\rho_S$ coefficients between these two variables and resolution for our dataset of 93,978 structures are 0.60 ($R_{work}$) and 0.64 ($R_{free}$)): lower resolution structures have higher $R$-factors as there are fewer experimental data available to which to fit the model, hence the model and data display the poorer agreement.

**$B_{net}$-percentile does not correlate with resolution.** In an attempt to overcome this issue, we extended our analysis to compare $B_{net}$ values of structures with similar resolutions. Many metrics of crystal structure quality are correlated with the resolution, hence they are commonly compared across subsets of structures of similar resolution. In our analysis, we used the same methodology as used by the PDB to compare structures of similar resolution: specifically, we calculated the percentile ranking of each structure's $B_{net}$ value in the subset of structures closest in resolution (first the resolution range encompassing the 1000 closest structures was identified, all structures falling in this range were then included in the subset—see Methods for more details).

When calculating Spearman's rank correlation coefficients between $B_{net}$-percentile and the eight variables listed above (Fig. 5 and Supplementary Table 5), we find that the correlation

coefficients of all eight variables with $B_{net}$-percentile fall in the range of $-0.2 < \rho_S < 0.2$, thus demonstrating that (barring influence from a variable we have overlooked) $B_{net}$-percentile values are predominantly affected by radiation dose, and correspondingly by damage.

This of course though does not mean that $B_{net}$ and $B_{net}$-percentile values are never affected by other variables. As an example, even though the correlation between $B$-factor restraint weights and $B_{net}$ is weak, if very strong $B$-factor restraints are applied to a structure this will prevent the $B$-factor values of damaged atoms from increasing relative to undamaged atoms, and hence will reduce $B_{net}$. However, whilst there will be a few exceptions, the at best weak correlations between $B_{net}$ and the eight selected variables mean such exceptions should be rare.

Consequently, based upon the analysis in this and the previous section, we are confident that $B_{net}$ values can be used to compare the extent of radiation damage suffered by structures of a similar resolution, and $B_{net}$-percentile values can be used to compare structures at different resolutions. Together these two metrics will provide users with a powerful tool to interrogate their own plus others' structures for radiation damage artefacts.

**$B_{net}$ and $B_{net}$-percentile identify damaged structures in the PDB.** To demonstrate the utility of these two metrics, we manually examined the ten structures with the highest $B_{net}$ values (all of which also have $B_{net}$-percentile values of greater than 0.999—see Supplementary Table 6) in our dataset of 93,978 structures for radiation damage artefacts (the selected structures are 5WUC[43], 5FXL[44], 5XQP[45], 3S8S, 3UX1[46], 1V70, 6Q5R[47], 3A07[48], 6BKL[49], and 2XMK[50]). None of these structures mention radiation damage in either the authors' remarks in the PDB entry or if recorded, their associated publications. However, eight of the ten structures (5WUC, 5FXL, 5XQP, 3S8S, 1V70, 6Q5R, 3A07, and 2XMK) show clear evidence of radiation damage to their aspartate/glutamate residues in their electron density maps, with blobs of negative difference density visible around the carboxyl groups of many of their aspartate and glutamate side chains. Furthermore, the two structures that contain disulfide bonds (5FXL and 3A07) also show clear damage to these bonds (Fig. 6 and Supplementary Fig. 4).

Additionally, the unusually high $B_{net}$ values of the two structures that do not show obvious signs of damage (3UX1 and 6BKL) can be readily explained. In spite of having an $R_{free}$

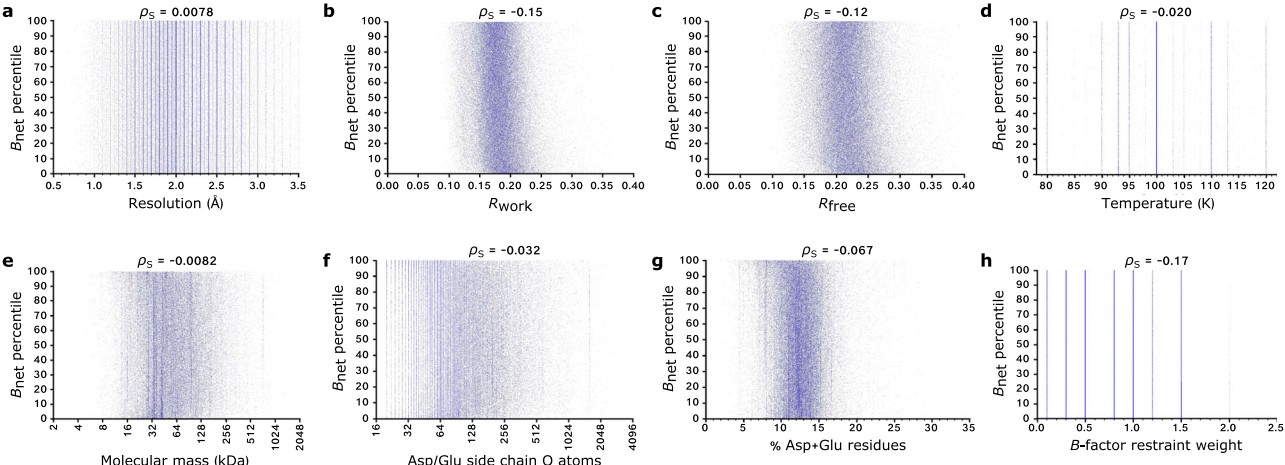

**Fig. 5 Investigating the dependence of $B_{net}$-percentile on various parameters.** Scatter plots of $B_{net}$-percentile vs.: **a** resolution; **b** $R_{work}$; **c** $R_{free}$; **d** temperature; **e** molecular mass; **f** the number of aspartate/glutamate side-chain oxygen atoms; **g** the percentage of aspartate/ glutamate residues; and **h** the strength of $B$-factor restraints applied in the final round of refinement. The Spearman's rank correlation coefficient ($\rho_S$) between the two variables is reported for each plot. Source data are provided at https://doi.org/10.5281/zenodo.5566558.

**Fig. 6 Observation of typical radiation damage features in structures with high $B_{net}$ values.** wwPDB validation statistics, models and density maps for the ten structures with the highest $B_{net}$ values in the PDB-REDO databank. For the eight structures with visible radiation damage artefacts in their associated electron density difference maps, a representative example is shown: for the two structures without such artefacts, the alternative causes of their high $B_{net}$ and $B_{net}$-percentile values are shown instead. $2mF_{obs} - DF_{calc}$ maps (blue) are contoured at 1.5 rmsd; $F_{obs} - F_{calc}$ difference density maps are contoured at ± 3.0 rmsd (green/red).

value of 0.28, the fit of model 3UX1 to the associated data is so bad that it is impossible to assess whether there are any damage artefacts present or not (Fig. 6). It is possible that the model's high $B_{net}$ value results from its aspartate and glutamate residues by chance displaying a worse fit to the density than the other residues in the structure, however, this hypothesis cannot be confirmed without extensive re-refinement of the structure.

Conversely, there is good agreement between model 6BKL and the data. However, the protein is an assembly of four identical copies of a 25-residue α-helical peptide. There are two aspartate residues per helix, one of which is two residues away from the N-terminus and the other of which is two residues away from the C-terminus. In general, the fit of a model to its corresponding data deteriorates at chain termini, and accordingly, atoms at these locations have higher $B$-factor values (which is indeed the case for 6BKL). Usually, the packing density values of the chain termini would be lower, and hence the $B_{Damage}$ metric would compensate for such increases in $B$-factor: however, in this case, because the structure is an oligomer of four short α-helices, the packing density does not vary much along the length of the helices. Consequently, the high $B_{net}$ value of 6BKL results from the close proximity of all of its aspartate residues to the chain termini, as opposed to radiation damage. There are very few other protein structures with such a distorted distribution of aspartate and glutamate residues throughout their chains, hence there will not be many other structures deposited in the PDB affected in this way. However, this result is a reminder that $B_{net}$ and $B_{net}$-percentile values need to be interpreted in the context of what is known about a structure. This is especially important for structures with high $B_{net}$ and $B_{net}$-percentile values that lie outside of the top 10: such structures are probably damaged, but are unlikely to display visible artefacts in their associated difference maps to enable "confirmation" of the $B_{net}$(-percentile) reading (hence the need to develop the metrics in the first place).

Strikingly, the wwPDB validation metrics reported for these 10 structures, with the exception of 3UX1, are reasonable to excellent (Fig. 6), and certainly do not reflect the large number of radiation damage artefacts afflicting them. Many users of the PDB are unaware of the possibility of radiation damage artefacts in deposited structures, moreover, most users only view the deposited model (without the associated electron density map). The $B_{net}$ and $B_{net}$-percentile metrics can help such users rapidly assess structure quality with respect to radiation damage, and thus help to prevent them from drawing incorrect biological conclusions from their own and others' structures. As an example, there are two models of actinohivin (an anti-HIV protein) in our dataset of 93,978: 3A07 and 4G1R. Both have good values across the standard wwPDB validation statistics, but whilst 3A07 has the eighth highest $B_{net}$ value in the entire dataset ($B_{net}$: 25.1 (3sf); $B_{net}$-percentile: 1), 4G1R is ranked 30,497th, and has much more reasonable-looking $B_{net}$ and $B_{net}$-percentile values ($B_{net}$: 1.61 (3sf); $B_{net}$-percentile: 0.489 (3sf)). 4G1R would thus be the better option of the two to take forwards for subsequent analysis.

## Discussion

Difficulties in detecting specific radiation damage artefacts mean that they are largely ignored when solving and analysing PX structures. However, these artefacts are not benign and can compromise the biological information drawn from a structure, for example as demonstrated by the X-ray-induced changes in the metal site conformation of photosystem II[6], and by the reduction of haem-containing proteins to redox states not relevant to their various mechanisms[51].

To address this problem, here we have developed the $B_{net}$ metric, which assesses the extent of radiation damage suffered by cryo-temperature (80–120 K) PX structures. To quantify

radiation damage, $B_{net}$ measures the fraction of and extent to which a structure's aspartate/glutamate side-chain oxygen atoms have higher $B_{Damage}$ values than the median $B_{Damage}$ value of all atoms in the structure. We have validated that $B_{net}$ can be fairly compared between different structures of similar resolution and developed the $B_{net}$-percentile metric to compare structures of different resolutions. Note that because we do not know the fraction of structures in the PDB that contain radiation damage artefacts, we have not attempted to set a threshold value for $B_{net}$ beyond which a structure is "damaged". Instead, it will be for the user to decide whether or not a structure's $B_{net}$ and $B_{net}$-percentile values will prevent them from using the structure, by taking into consideration other available measures of crystal quality plus the purposes for which they intend to use the structure. Nevertheless, to aid the user, as a rough guide we suggest that a $B_{net}$ value greater than 3.0, and/or a $B_{net}$-percentile value greater than 0.95, would merit further inspection of a structure.

Interestingly, we observe that the extent of radiation damage in the PDB has remained relatively constant over time, as measured by calculating the Spearman's rank correlation coefficient between $B_{net}/B_{net}$-percentile and deposition year (Supplementary Fig. 5). By flagging probable damaged structures for further inspection, we hope that the $B_{net}$ and $B_{net}$-percentile metrics will help users when inspecting a structure/deciding which structural model of a protein of interest they want to select for their analysis. We also hope that the metrics will help crystallographers to identify damaged structures before/during the PDB deposition process, and thus reduce the fraction of damaged entries in the PDB. Because they are $B$-factor-derived metrics, $B_{net}$ and $B_{net}$-percentile values should only be calculated for fully refined structures, so unfortunately they cannot be used to assess damage during the refinement process—however, they could be used to assess whether a structure should be re-refined with a truncated dataset.

In the future, when more data are available for validation, we hope to develop equivalents of these two metrics for structures containing nucleic acids, and for those derived from room temperature data. Currently, users can calculate $B_{net}$ values for PX structures by running the *RABDAM* software[52,53], which is distributed as part of the CCP4 software suite[54] or alternatively is available from https://github.com/GarmanGroup/RABDAM. $B_{net}$-percentile values for the dataset of 93,978 structures analysed in this publication are listed at https://doi.org/10.5281/zenodo.5566558.

## Methods

**$B_{Damage}$.** The $B_{Damage}$ value of an atom $j$ is calculated as the ratio of its $B$-factor to the arithmetic mean of the $B$-factor values of atoms 1 to $n$ that are classified as occupying a similar packing density environment (Eq. 2 and Fig. 1), with an atom's packing density being defined as the number of non-hydrogen atoms within a 7 Å radius in the context of the crystalline structure.

$$B_{Damage\,j} = \frac{B-factor_j}{\frac{1}{n}\sum_{i=1}^{i=n} B-factor_i} \quad (2)$$

**$B_{net}$ calculation.** $B_{net}$ is calculated from the kernel density estimate (KDE) of the $B_{Damage}$ values of a structure's aspartate and glutamate side-chain oxygen atoms, as the ratio of the area under the curve on either side of the median $B_{Damage}$ value of all atoms in the structure. The median was selected as the measure of central tendency because the value of the mean is susceptible to distortion by outliers, whilst the modal value is much more sensitive to the shape of the distribution (which can be irregular and/or noisy in the case of smaller structures). The KDE is plotted in Python using a Gaussian kernel, whilst the kernel bandwidth is set using the Scott method[55]. The area under the KDE is measured using the trapezium rule, split into 99 trapeziums spanning equal intervals along the $B_{Damage}$ (x-)axis. The trapezium into whose area the median $B_{Damage}$ value of all atoms in the structure falls is included in the below-median area (area A in Eq. 1) calculation. The code for calculating $B_{net}$ is included in the *RABDAM* software[52], which is open-source and available from GitHub (https://github.com/GarmanGroup/RABDAM). *RABDAM* is also distributed as part of the CCP4 software suite[54].

Four different measures of skewness were calculated in addition to the $B_{net}$ metric for the structures derived from the 23 radiation damage series analysed (see

"Radiation dataset analysis" section below), and lines of best fit between each metric and dose were calculated using least squares regression for all 23 series. When considering the goodness of fit (measured by $R^2$) of these lines of best fit, plus the consistency of their gradients and y-intercepts (measured by normalised standard deviation), $B_{net}$ was judged to achieve the best performance of the five options. The results of this analysis are available at https://doi.org/10.5281/zenodo.5567741.

**Radiation dataset analysis**. Radiation damage datasets were identified by searching the PDBe (as of 19 November 2020) for "radiation damage", "serial crystallography" and "dose", then manually filtering the results to select structures in radiation damage series. Each selected radiation damage series: (i) is of a protein structure (without a nucleic acid component); (ii) was collected at a temperature in the range of 80–120 K; (iii) contains three or more models; (iv) every underlying dataset in the series has been collected from the same crystal(s); (v) has the dose (reported in Gy) absorbed by each structure derived from the series recorded in its associated publication; (vi) has no aspartate or glutamate residues whose total occupancy across all conformers is less than one; (vii) has been refined with per-atom $B$-factors (either isotropic or anisotropic); (viii) has been solved to a resolution of 3.5 Å or better; and (ix) has a $R_{free}$ value less than or equal to 0.4. Note that although occupancy refinement would be the correct treatment for decarboxylated side chains, this is rarely performed. This is because, as described in the main text, specific radiation damage artefacts are difficult to identify in individual PX structures, hence in most cases users are unaware of the damage in their structures. In order to detect damage, the $B_{Damage}$ and $B_{net}$ metrics exploit the increased $B$-factor values that result when damaged, sub-1 occupancy side chains are refined as full occupancy.

Metadata describing these structures were taken from their associated mmCIF files. The two exceptions to this were: (i) the Wilson $B$-factor, which was taken from the associated PDB validation report; and (ii) the dose, which had to be manually extracted from the relevant publication. All Wilson $B$-factor values were calculated using Xtriage. The doses of all structures were calculated using RADDOSE (versions 1 or 2[56,57] or -3D[3,58]), although the use of RADDOSE was not a prerequisite for selection. The 23 datasets and their associated metadata are listed in Radiation_damage_datasets_SI.csv, which is available at https://doi.org/10.5281/zenodo.5566557. The script used to extract the metadata for each structure in the dataset is available at https://github.com/kls93/Bnet_PDB_parsing.

$B_{net}$(Asn/Gln) values were obtained by calculating the ratio of the area on either side of the median $B_{Damage}$ value of all atoms in the structure under a KDE of the $B_{Damage}$ values of that structure's asparagine and glutamine side-chain oxygen atoms.

The four radiation damage datasets listed in Supplementary Table 4, selected to analyse the effects on their structures' $B_{net}$ values of changing the $B$-factor restraint weights applied during the refinement, were re-refined by two different methods. Firstly, the structures were downloaded from the PDB-REDO databank; and secondly, the (original PDB) structures were subjected to three macrocycles of unrestrained $B$-factor refinement (whilst the model coordinates were constrained to remain constant) with the refinement package in the Phenix software suite[59]. Note that because we do not know the $B$-factor restraint weights applied to the original (PDB) models, there is uncertainty that either method reduces the $B$-factor restraint weights of the re-refined structures below those applied to the original structures. However, it is highly unlikely that any of the original structures were subjected to unrestrained $B$-factor refinement. Moreover, we can be confident that the PDB-REDO structures have not been refined with extreme $B$-factor restraints.

**Generating a dataset of PX structures**. A dataset of PX structures was obtained by filtering the PDB using the following criteria: (1) the entry was released on or before 19 November 2020; (2) it was solved by X-ray diffraction; (3) the structure is a protein; (4) it does not contain any nucleic acid components; and (5) experimental data are available. This generated a dataset of 120,311 structures. We then filtered this dataset to retain only those structures containing one or more aspartate/glutamate residues (either L- or D-isomers). The resultant dataset contained 120,075 structures, i.e. 99.8% (3sf) of the original 120,311. Further filtering to retain those structures collected at a temperature in the range of 80–120 K reduced the dataset to 105,856 structures, i.e. 88.0% (3sf) of the original 120,311.

**PDB-REDO analysis**. We searched the PDB-REDO databank[42] (as of 19 November 2020) for PX structures that met all of the required criteria for the radiation damage datasets (see "Radiation dataset analysis" methods), excluding the need for the dose to be recorded or for them to be part of a radiation damage dataset giving three or more structures. We also applied filters to select structures that (i) contained 20 or more aspartate/glutamate side-chain oxygen atoms, and (ii) have not been refined with a flat $B$-factor model (information about the latter is not available for PDB structures, hence this criterion was not applied to the radiation damage datasets). Of the 105,856 cryo-temperature (80–120 K) PX structures in our PX dataset described above, 93,978 (88.8% (3sf)) met these criteria and were successfully downloaded from the PDB-REDO databank, processed by the RABDAM software to calculate $B_{net}$, and had values recorded for each of the eight variables whose correlation coefficient with $B_{net}$ we assessed. As described in the main text, we calculated Spearman's rank correlation coefficients of $B_{net}$ with resolution, $R_{work}$, $R_{free}$, temperature, molecular mass, the percentage and the raw number of aspartate/glutamate residues a structure contains,

and the strength of the $B$-factor restraints applied in the final round of refinement of a structure. The values of all these features, including $B_{net}$, were measured for the asymmetric unit. Molecular mass, plus numbers of aspartate and glutamate residues, were calculated for all atoms expected to be present in the asymmetric unit (i.e. including unmodelled atoms). In contrast, because $B_{net}$ is a $B$-factor-derived metric, its value is calculated from the modelled atoms only. All metadata associated with each PDB-REDO entry were taken from the corresponding mmCIF file (we used "XXXX_final.cif", to ensure that total $B$-factors are reported in the isotropic $B$-factor records as required by RABDAM). The exceptions to this were the $B$-factor restraint weights and the Wilson $B$-factor, which were taken from the associated data.json file for the PDB-REDO entry. Wilson $B$-factors for PDB-REDO entries have been calculated with SFcheck.

$B_{net}$-percentile values were calculated by first determining the resolution range encompassing the 1000 structures out of the dataset of 93,978 that were closest in resolution; the percentile value of the structure's $B_{net}$ score was then calculated relative to all structures in the dataset whose resolution lay in the calculated resolution range. Spearman's rank correlation coefficients were also calculated between $B_{net}$-percentile and the same eight variables as considered for $B_{net}$. In addition, we calculated Spearman's rank correlation coefficients of both $B_{net}$ and $B_{net}$-percentile with the year of deposition in the PDB (which was retrieved from the mmCIF file for the PDB entry). The script used to determine values for these eight variables for every structure in the dataset, plus for calculating their $B_{net}$-percentile values and finding the year of deposition in the PDB, is available at https://github.com/kls93/Bnet_PDB_parsing, whilst the output data frame from running this script is provided in csv format at https://doi.org/10.5281/zenodo.5566558.

Finally, we looked for radiation damage artefacts in the associated electron density maps of the ten structures with the highest $B_{net}$ values out of our dataset of 93,978 PDB-REDO structures. However, the wwPDB validation statistics and, where relevant, the models and electron density maps showing radiation damage artefacts, as presented in Fig. 6, are for the corresponding PDB structures. This is because the wwPDB validation metrics (both raw and percentile values) are those that users most commonly utilise to assess and compare crystal structure quality, and these metrics are not similarly presented (or calculated at all in the case of the percentile metrics) in the PDB-REDO databank. For comparison, the $B_{net}$ values of these corresponding PDB structures are listed in Supplementary Table 6—in all cases, their $B_{net}$ values are also unusually high (higher than the $B_{net}$ values measured for even the most damaged structures in the 23 radiation damage series).

## Data availability

$B_{net}$ and $B_{net}$-percentile values for the dataset of 93,978 PDB-REDO structures, and $B_{net}$ values for the 23 radiation damage datasets, have been deposited in the Zenodo repository at https://doi.org/10.5281/zenodo.5566558 and https://doi.org/10.5281/zenodo.5566557, respectively. A comparison of the performance of four different skewness metrics to the $B_{net}$ metric has been deposited in the Zenodo repository at https://doi.org/10.5281/zenodo.5567741. The accession codes of all 93,978 PDB codes used in this study are listed at https://doi.org/10.5281/zenodo.5566558 and can be accessed via their four-digit accession codes in the Protein Data Bank repository (https://www.rcsb.org/). Source data are provided with this paper.

## Code availability

All of the code used to perform the above analysis is open-source and available from GitHub. The $B_{net}$ metric is calculated by the RABDAM software, which is available at https://github.com/GarmanGroup/RABDAM and https://zenodo.org/record/5932846. The script used to extract metadata (including values for each of the eight variables whose correlation with $B_{net}$ is tested) for each PDB-REDO structure, plus for calculating $B_{net}$-percentile values, is available on GitHub at https://github.com/kls93/Bnet_PDB_parsing. The script used to extract metadata for the PDB structures included in the analysis of 23 radiation damage series is also available at this location.

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

## Acknowledgements

We gratefully acknowledge the Moritz-Heyman Scholarship Program and the BBSRC (Biotechnology and Biological Sciences Research Council) for funding summer internships for K.L.S. to complete this work. We thank Edward Lowe for helping to run *RABDAM* on the PDB-REDO databank, and for useful discussions concerning *B*-factors and *B*-factor restraints. We also thank, Ian Carmichael, Joshua Dickerson and Markus Gerstel for constructive comments on this manuscript.

## Author contributions

K.L.S. and E.F.G. conceived the project. K.L.S. designed and performed the analysis. K.L.S. and E.F.G. wrote the manuscript.

## Competing interests

The authors declare no competing interests.
