## [Peer Review File · Nature Communications]

REVIEWER COMMENTS

Reviewer #1 (Remarks to the Author):

The authors present an interesting and (to my knowledge) unique metric to assess overall radiation damage in a way that can be used in model validation and dataset generation. Together with Bdamage there is now a way to see which structures should be interpreted with care and which parts of the model.

The manuscript is very clear and the authors have made sure that their developments are immediately available. The data is also available albeit that in future studies it should be stored in a more FAIR format such as CSV or JSON. I think only a few changes are needed for the manuscript to be accepted.

- The title could be improved, especially because a lot of the analysis is actually not done on models directly from the PDB. Perhaps highlight the fact that there is now a metric that can be used to compare PDB entries on equal footing.
- Page 2, 1st par: "light sources" have "resulted"
- Page 2, 3rd par: I think "liberated" is not quite the right word here. I'm not sure the electrons want to be free, but we don't want them to be. Perhaps "escaped" is a better term or more neutral something like "emitted/ejected/released".
- Equation 2: it is not immediately clear what LHS and RHS stand for.
- Page 10: I agree with leaving out low resolution cases, but the statement about not having enough data to support the refinement without additional restraints at 3.0Å makes little sense here. Even at 2.0Å resolution we need restraints for x, y, z, B . At lower resolution we just tighten restraints or we change the model parameterization to reduce the number of independent parameters (e.g. by grouping B-factors). Nothing magic happens at 3.0Å resolution. Please rephrase.
- Figure 3: the symbols for the data points are a bit too small to tell apart.
- Page 14: Which program was used to calculate the Wilson B-factor?
- Page 14: the word "bar" makes the last sentence harder to read for non-native speakers than necessary.
- Page 16: please rephrase the last sentence as R-factors are not directly related to electron density.
- Figure 4: if there are any points with R-factor/R-free > 0.4 in the dataset they should be removed. Either the original PDB model was already quite terrible or PDB-REDO made it so. This would allow the x-axis for 4B, 4C, to be changed so that the plot becomes more informative. Same for 4G.

- Page 20: perhaps the authors can give an example of a case where they would prefer one PDB entry over another for analysis based on Bnet(-percentile). There must be homologous PDB(-REDO) entries with wildly diverging Bnet values.
- Discussion: To what extent could Bnet be used as a post-hoc analysis of data processing? With the advent of having unmerged reflection data in the PDB it could be interesting to reject later observations if the data multiplicity allows it.
- Discussion: Putting Bnet(-percentiles) into a historical perspective could be interesting. Has the amount of radiation damage for PDB entries changed over the years?
- Discussion: the authors show in table S4 that the refinement protocol can affect the perceived Bnet value. It would be interesting to see this for a broader dataset. Is there a trend between the calculated Bnet in PDB models and their PDB-REDO counterparts? Table S6 suggest that the Bnet values in te PDB might be lower, but that would be based on too few observations.
- Methods, general: The authors should be careful when to use the term structure, (structure) model, or something else. E.g. a radiation damage series...contains three or more models; iv) every underlying dataset in the series has been collected...
- Page 23: remove “(“ before “Note”
- Page 23: which mmCIF key is used to extract radiation damage?
- Page 23: structure coordinates -> model coordinates or atomic coordinates
- Page 23: “1) the structure was released...” -> “1) the entry was released...”
- Page 24: ...and ii) have NOT been refined with a flat...
- Page 24: is the molecular mass taken from the coordinates or from the deposited sequence? And was the mass of a chain used or of the ASU or something else?
- Page 24: For figure 6 the authors can use the PDB validation server to make sliders of the PDB-REDO models.
- References: <https://doi.org/10.1002/pro.3353> seems to be a more modern reference of the PDB-REDO databank
- References: It makes more sense to number the supplemental references continuing from the main references.

Reviewer #2 (Remarks to the Author):

This manuscript presents what appears to be an effective and general method of detecting telltale signs of radiation damage in protein structures, even if the authors of those structures did not. The result is timely and should be very useful. I recommend publication in this journal.

Shortcomings of the work as submitted lie in a few minor stylistic issues, and perhaps one overarching lack of clarity as to why the analysis was done the way it was. The shortcomings of alternative statistical approaches are claimed, but little to no evidence is presented to justify these claims. In general, these claims should either be removed in favor of a briefer description of what was done, or a very short description of what went wrong when/if alternative methods were applied.

Central to all this is the use of kernel density estimation (KDE). This is a rather convoluted approach without much historical precedence. It also has the disadvantage of having hidden tuning variables, such as the kernel choice and bandwidth estimation. Did the authors explore any other more traditional estimators of skewness? The text mentions a few of them, such as Pearson's moments and quartile-based estimates, but it is not clear from the text if these alternate analyses were carried out and found inferior to the kernel density estimates, or if the authors simply expected KDE would be superior and never attempted more traditional alternatives. This effort, if it was made, is valuable to mention here. I note the statement:

"this area ratio is more strongly influenced by the number of BDamage data points either side of the median,"

implies that the number of BDamage data points on either side of the median might be a superior metric to the KDE approach. Was such an analysis carried out? Or is this a conjecture?

Abstract:

"numerous damaged structures,"

please replace with the actual number

Intro:

"increasing flux densities of synchrotron light sources has resulted in radiation damage"

Although this statement is a time-honored boiler-plate for radiation damage literature, is it really the flux density that is to blame? Or is it rather a poor understanding of the consequences of too-long exposures combined with the much more rapid feedback on the most desirable metric (resolution)?

Results:

The description of the method doesn't belong in the Results section

"potential measure"

change to past tense, this is a report, not a proposal

"would be suboptimal"

change to past tense

"it is necessary to focus upon only one type of damage event"

were other approaches tried? or is this a supposition/expectation?

"absorbed dose"

can always be shortened to

"dose"

"resolution of less than 3.5"

is ambiguous since numerically smaller resolution values are not what is meant. Perhaps better as:

"resolution of worse than 3.5"

"It is widely accepted that"

please cite a recent review

"considerable error associated with dose calculations,"

it may be worth mentioning that

"automatically re-refined"

is confusing. Was re-refinement triggered by the low gradient? Perhaps better as:

"re-refined"

Fig 6: why are the map images missing from 5xqp, 3ux1 and 6bkl ? You must know which ones are the "worst" ?

NCOMMS-21-24783, REVIEWER COMMENTS 23-8-21

Shelley and Garman

Our responses are in red below.

Reviewer #1 (Remarks to the Author):

The authors present an interesting and (to my knowledge) unique metric to assess overall radiation damage in a way that can be used in model validation and dataset generation. Together with B_{damage} there is now a way to see which structures should be interpreted with care and which parts of the model.

The manuscript is very clear and the authors have made sure that their developments are immediately available. The data is also available albeit that in future studies it should be stored in a more FAIR format such as CSV or JSON. I think only a few changes are needed for the manuscript can be accepted.

We thank the reviewer for these positive comments.

We have resubmitted the data as csv files: see <https://doi.org/10.5281/zenodo.5566557> and <https://doi.org/10.5281/zenodo.5566558>.

- The title could be improved, especially because a lot of the analysis is actually not done on models directly from the PDB. Perhaps highlight the fact that this is now a metric that can be used to compare PDB entries on equal footing.

We agree and have changed the title to “Quantifying and comparing radiation damage in the Protein Data Bank”

- Page 2, 1st par: “light sources” have “resulted”
Changed “has” to “have”

- Page 2, 3rd par: I think “liberated” is not quite the right word here. I’m not sure the electrons want to be free, but we don’t want them to be. Perhaps “escaped” is a better term or more neutral something like “emitted/ejected/released”.

We liked this comment! Changed “liberated” to “ejected”.

- Equation 2: it is not immediately clear what LHS and RHS stand for.

Apologies. Changed “LHS” and “RHS” in figure 2 and equation 2 to “A” and “B” respectively, and defined their meaning in the text by adding:

“A and B are the areas under the KDE estimate of the $B_{\text{Damage-Asp/Glu}}$ distribution to the left- and right-hand side of the median B_{Damage} value of all atoms in the structure, respectively.”

- Page 10: I agree with leaving out low resolution cases, but the statement about not having enough data to support the refinement without additional restraints at 3.0Å makes little sense here. Even at 2.0Å resolution we need restraints for x,y,z,B. At lower resolution we just tighten restraints or we change the model parameterization to reduce the number of independent parameters (e.g. by grouping B-factors). Nothing magic happens at 3.0Å resolution. Please rephrase.

Agreed: our original phrasing makes 3.0 Å sound like a threshold, which was not our intention. We have rephrased this as:

“Additionally, we excluded datasets containing one or more structures with a resolution worse than 3.5 Å; as resolution deteriorates, the number of observed reflections decreases, meaning fewer parameters can be refined. It is highly unlikely that per-atom B-factor refinement is the appropriate choice for structures solved to worse than 3.5 Å resolution. Lastly, we removed datasets containing one or more structures with an R_{free} value greater than 0.4, since these models show such poor fit to the electron density data that detection of radiation damage artefacts is very difficult if not

impossible.”

- Figure 3: the symbols for the data points are a bit too small to tell apart.

We have made them bigger. Note that none of the datasets on the same graph is the same colour – the symbol differences are included to make the datasets more readily distinguishable, but they are not needed to tell the difference.

- Page 14: Which program was used to calculate the Wilson B-factor?

For the analysis of the PDB structures in the radiation damage series, the Wilson B-factor was taken from the value reported in the “Full Report” available from the PDB website, which is calculated using Xtrriage. We have made it clear in the methods that Xtrriage is the program used to calculate these values, to ensure that readers do not think the variation between datasets could have been caused by the depositors using different programs to calculate Wilson B for their structures.

For the PDB-REDO structures, Wilson B-factor was taken from the “BWILS” record in the data.json file associated with each entry. SFcheck is used to calculate these Wilson B-factor values. Again, we have now made this clear in the Methods by adding:

“The two exceptions to this were: i) the Wilson B-factor, which was taken from the associated PDB validation report; and ii) the dose, which had to be manually looked up in the relevant publication. All Wilson B-factor values were calculated using Xtrriage.”

and

“All metadata associated with each PDB-REDO entry were taken from the mmCIF file (“XXXX_final.cif”, to ensure that total *B*-factors are reported in the isotropic *B*-factor records as required by *RABDAM*). The exceptions to this were the *B*-factor restraint weights and the Wilson *B*-factor, which were taken from the associated data.json file for the PDB-REDO entry. Wilson *B*-factors for PDB-REDO entries have been calculated with SFcheck.”

- Page 14: the word “bar” makes the last sentence harder to read for non-native speakers than necessary.

We changed all instances of “bar” in the text to either “apart from” or “except”.

- Page 16: please rephrase the last sentence as R-factors are not directly related to electron density. Changed to “lower resolution structures have higher R-factors as there are fewer experimental data available to which to fit the model, hence the model and data display poorer agreement”.

- Figure 4: if there are any points with R-factor/R-free > 0.4 in the dataset they should be removed. Either the original PDB model was already quite terrible or PDB-REDO made it so. This would allow the x-axis for 4B, 4C, to be changed so that the plot becomes more informative. Same for 4G. Agreed, so we have taken out structures with $R_{\text{free}} > 0.4$. This has reduced the size of the dataset from 94,145 structures to 93,978, plus caused minor changes in some of the smaller correlation coefficient values.

- Page 20: perhaps the authors can give an example of a case where they would prefer one PDB entry over another for analysis based on B_{net} (-percentile). There must be homologous PDB(-REDO) entries with wildly diverging B_{net} values.

We have updated the text to mention that there is another structure of the protein in entry 3A07 (one of the 10 structures with highest B_{net} value in our dataset, but whose stats otherwise look pretty good) that has similarly good stats but a much more reasonable B_{net} value.

It reads:

“As an example, there are two models of actinohivin (an anti-HIV protein) in our dataset of 93,978: 3A07 and 4G1R. Both have good values across the standard wwPDB validation statistics, but whilst 3A07 has the eighth highest B_{net} value in the entire dataset (B_{net} : 25.1 (3sf); B_{net} -percentile: 1), 4G1R

is ranked 30,497th, and has much more reasonable-looking B_{net} and B_{net} -percentile values (B_{net} : 1.61 (3sf); B_{net} -percentile: 0.489 (3sf)). 4G1R would thus be the better option of the two to take forwards for subsequent analysis.”

- Discussion: To what extent could Bnet be used as a post-hoc analysis of data processing? With the advent of having unmerged reflection data in the PDB it could be interesting to reject later observations if the data multiplicity allows it.

This is possible, however as it is designed to be used on fully-refined, deposition-ready structures, it would be quite a lot of work for the user. We have now mentioned this in the discussion:

“Because they are B-factor-derived metrics, B_{net} and B_{net} -percentile values should only be calculated for fully refined structures, so unfortunately they cannot be used to assess damage during the refinement process – however, they could be used to assess whether a structure should be re-refined with a truncated dataset.”

- Discussion: Putting Bnet(-percentiles) into a historical perspective could be interesting. Has the amount of radiation damage for PDB entries changed over the years?

We have run this analysis, and there is no change. We have included new plots as Figure S4, which is mentioned in the discussion:

” Interestingly, we observe that the extent of radiation damage in the PDB has remained relatively constant over time, as measured by calculating the Spearman’s rank correlation coefficient between B_{net}/B_{net} -percentile and deposition year (Fig. S4).”

- Discussion: the authors show in table S4 that the refinement protocol can affect the perceived Bnet value. It would be interesting to see this for a broader dataset. Is there a trend between the calculated Bnet in PDB models and their PDB-REDO counterparts? Table S6 suggest that the Bnet values in the PDB might be lower, but that would be based on too few observations.

In general there is good agreement between the PDB and PDB-REDO results. For extreme examples there will be differences as PDB-REDO and/or the authors of the PDB deposition make mistakes during the refinement. As we are selecting the 10 structures in PDB-REDO with the highest Bnet values, it is therefore not surprising that the PDB Bnet values are slightly lower.

To demonstrate that this is not usually the case though, we have plotted B_{net} for PDB vs. PDB-REDO structures for the 23 radiation damage datasets. There is generally good agreement between the values, and when we discard one extreme outlier, we obtain a line of best fit of gradient 0.95 (2sf) and y-intercept -0.043 (2sf), which demonstrates good agreement (would expect the values to be 1 and 0, respectively, if Bnet values for equivalent PDB and PDB-REDO structures were identical).

Below is an image of the plot – we have not included it as a supplementary figure in the paper as we are not sure this is necessary.

- Methods, general: The authors should be careful when to use the term structure, (structure) model, or something else. E.g. a radiation damage series...contains three or more models; iv) every underlying dataset in the series has been collected...

Thank you for pointing this out. We have tried to clarify these instances throughout the manuscript.

- Page 23: remove “(“ before “Note”

Removed.

- Page 23: which mmCIF key is used to extract radiation damage?

Dose is not recorded in PDB records. The values had to be manually found from the associated publications. We have now made this clearer in the methods text.

“The two exceptions to this were: i) the Wilson *B*-factor, which was taken from the associated PDB validation report; and ii) the dose, which had to be manually looked up in the relevant publication.”

- Page 23: structure coordinates -> model coordinates or atomic coordinates

Changed to “model coordinates” as suggested.

- Page 23: “1) the structure was released...” -> “1) the entry was released...”

Changed as suggested.

- Page 24: ...and ii) have NOT been refined with a flat...

Thank you for spotting this omission! Much appreciated.

- Page 24: is the molecular mass taken from the coordinates or from the deposited sequence? And was the mass of a chain used or of the ASU or something else?

It is taken from the asymmetric unit (mmCIF record `_pdbx_poly_seq_scheme`). This has now been made clearer in the methods text.

“The values of all these features, including *B*_{net}, were measured for the asymmetric unit. Molecular mass, plus numbers of aspartate and glutamate residues, were calculated for all atoms expected to be present in the asymmetric unit (i.e. including unmodelled atoms).”

- Page 24: For figure 6 the authors can use the PDB validation server to make sliders of the PDB-REDO models.

This is a good suggestion. However, we would like the metrics we show in the figure to match up with those that readers will see in the PDB if they decide to look them up, so we would like to leave these as they are.

- References: <https://doi.org/10.1002/pro.3353> seems to be a more modern reference of the PDB-REDO databank

We have replaced reference 42 with this newer reference.

- References: It makes more sense to number the supplemental references continuing from the main references.

All the radiation damage series are referenced in the main text (references 17,19-35) and have now been relinked so that Table S1 has the references as referred to in the main text but in sequential order. We hope this is what the referee meant. If preferred, at the end of the Supplementary Information we could repeat the relevant references with the same numbers as in the main text.

Reviewer #2 (Remarks to the Author):

This manuscript presents what appears to be an effective and general method of detecting tell tale

signs of radiation damage in protein structures, even if the authors of those structures did not. The result is timely and should be very useful. I recommend publication in this journal.

We thank the reviewer for these positive comments.

Shortcomings of the work as submitted lie in a few minor stylistic issues, and perhaps one overarching lack of clarity as to why the analysis was done the way it was. The shortcomings of alternative statistical approaches are claimed, but little to no evidence is presented to justify these claims. In general, these claims should either be removed in favor of a briefer description of what was done, or a very short description of what went wrong when/if alternative methods were applied.

Central to all this is the use of kernel density estimation (KDE). This is a rather convoluted approach without much historical precedence. It also has the disadvantage of having hidden tuning variables, such as the kernel choice and bandwidth estimation. Did the authors explore any other more traditional estimators of skewness? The text mentions a few of them, such as Pearson's moments and quartile-based estimates, but it is not clear from the text if these alternate analyses were carried out and found inferior to the kernel density estimates, or if the authors simply expected KDE would be superior and never attempted more traditional alternatives. This effort, if it was made, is valuable to mention here. I note the statement:

"this area ratio is more strongly influenced by the number of BDamage data points either side of the median,"

implies that the number of BDamage data points on either side of the median might be a superior metric to the KDE approach. Was such an analysis carried out? Or is this a conjecture?

This analysis was carried out when we first developed the metric (the method we have chosen is more complicated to code up, so we wanted to try simpler methods first!). We judged our KDE metric to perform best of the range of metrics we tried. We have now deposited this analysis at <https://doi.org/10.5281/zenodo.5567741>, and updated the main text and methods to describe this analysis. In particular, in the Methods we have added:

"Four different measures of skewness were calculated in addition to the B_{net} metric for the structures derived from the 23 radiation damage series analysed (see "Radiation dataset analysis" section below), and lines of best fit between each metric and dose were calculated using least squares regression for all 23 series. When considering the goodness of fit (measured by R^2) of these lines of best fit, plus the consistency of their gradients and y-intercepts (measured by normalised standard deviation), B_{net} was judged to achieve the best performance of the five options. The results of this analysis are available at <https://doi.org/10.5281/zenodo.5567741>."

Abstract:

"numerous damaged structures,"

please replace with the actual number

We're hesitant to set a threshold for B_{net} for the reasons we mention in the discussion: we don't know the fraction of damaged structures in the PDB, since the damage is often difficult to distinguish from e.g. side chain disorder. Hence we would prefer to not provide a specific value here.

Intro:

"increasing flux densities of synchrotron light sources has resulted in radiation damage"

Although this statement is a time-honored boiler-plate for radiation damage literature, is it really the flux density that is to blame? Or is it rather a poor understanding of the consequences of too-long exposures combined with the much more rapid feedback on the most desirable metric (resolution)?

The referee is completely correct! We have changed the text to clarify the point and it now reads:

"..... data collection methodologies using the increasing flux densities of synchrotron light sources have resulted in radiation damage remaining one of the major challenges in protein

crystallography."

Results:

The description of the method doesn't belong in the Results section

We have moved some of the description of the method to the methods section (namely the description of the BDamage metric, as this has previously been defined in Gerstel et al., 2015). However, we have kept the description of how the Bnet metric works in the main text, for since this is a methods paper, the method is part of the result!

"potential measure"

change to past tense, this is a report, not a proposal

Changed from conditional to present tense, now reads "can thus be used as a measure".

"would be suboptimal"

change to past tense

Changed from conditional to present tense, now reads "is suboptimal".

"it is necessary to focus upon only one type of damage event"

were other approaches tried? or is this a supposition/expectation?

This is a really good question. We did try including disulfides when we first developed the metric. Their inclusion had negligible effect upon performance. However, when we thought about the metric in more detail, it became clear that since different structures contain different relative proportions of disulfides and Asp/Glu, and the two damage at different rates, it is not fair to include both. For instance, if you consider two structures with equally damaged Asp/Glu, but the second also has disulfide bonds (which will be damaged to a greater extent than the Asp/Glu residues), the second would have a higher B_{net} value if we were to include Asp/Glu and disulfides in our metric, whereas we want the structures to have the same value. We want them to have the same value because otherwise it would create difficulties when comparing structures with features such as certain cofactors and metal ions that we know are easily damaged, but which we cannot include in our metric (since we can't calculate BDamage values for non-protein atoms). Thus in practice, including disulfides does not have much effect, although from a theoretical viewpoint it was decided to exclude them. To clarify this, in the text we have changed "it is necessary" to "we decided to".

"absorbed dose"

can always be shortened to

"dose"

Changed throughout the manuscript.

"resolution of less than 3.5"

is ambiguous since numerically smaller resolution values are not what is meant. Perhaps better as:

"resolution of worse than 3.5"

Agreed, so we have changed all instances of this.

"It is widely accepted that"

please cite a recent review

We have added a citation to JM Holton JSR 2009.

"considerable error associated with dose calculations,"

it may be worth mentioning that

This comment appears to have been cut short, so we are not sure what is required here.

"automatically re-refined"

is confusing. Was re-refinement triggered by the low gradient? Perhaps better as:

"re-refined"

Series for re-refinement were selected manually, so we meant to say that no manual model building was carried out during the re-refinement. We agree that the sentence could be read with another interpretation and could be confusing. We have removed "automatically", and directed the user to the methods section for a description of the refinement performed.

Fig 6: why are the map images missing from 5xqp, 3ux1 and 6bkl ? You must know which ones are the "worst" ?

We have looked again at 5XQP, and seen that we were being very conservative when excluding a map image of it. It does contain signs of radiation damage (but just not around "a majority" of Asp/Glu side chains, which was our original criterion for including a map image in the manuscript). The fit of 3UX1 model to the data is very poor indeed, so it is impossible to say whether it is damaged or not.

6BKL does not contain any obvious radiation damage artefacts in the difference density map. It could still be damaged, but the more likely cause for its high B_{net} value is that its Asp residues are all located at the chain termini (it is a homotetramer of a 25 residue peptide), and the fit of the model to the density at the chain termini is worse than for the rest of the structure. Again, we have included an image to demonstrate this.

We have thus added map images to the empty boxes, and also described the issues with 3UX1 and 6BKL in detail in the main text by adding:

"Additionally, the unusually high B_{net} values of the two structures that do not show obvious signs of damage (3UX1 and 6BKL) can be readily explained. In spite of having an R_{free} value of 0.28, the fit of model 3UX1 to the associated data is so bad that it is impossible to assess whether there are any damage artefacts present or not (Fig. 6). It is possible that the model's high B_{net} value results from its aspartate and glutamate residues by chance displaying a worse fit to the density than the other residues in the structure, however this hypothesis cannot be confirmed without extensive re-refinement of the structure. Conversely, there is good agreement between model 6BKL and the data. However, the protein is an assembly of four identical copies of a 25-residue α -helical peptide. There are two aspartate residues per helix, one of which is two residues away from the N-terminus and the other of which is two residues away from the C-terminus. In general, the fit of a model to its corresponding data deteriorates at chain termini, and accordingly atoms at these locations have higher B-factor values (which is indeed the case for 6BKL). Usually the packing density values of the chain termini would be lower, and hence the B_{Damage} metric would compensate for such increases in B-factor: however, in this case because the structure is an oligomer of four short α -helices, the packing density does not vary much along the length of the helices. Consequently, the high B_{net} value of 6BKL results from the close proximity of all of its aspartate residues to the chain termini, as opposed to radiation damage. There are very few other protein structures with such a distorted distribution of aspartate and glutamate residues throughout their chains, hence there will not be many other structures deposited in the PDB affected in this way. However, this result is a reminder that B_{net} and B_{net} -percentile values need to be interpreted in the context of what is known about a structure. This is especially important for structures with high B_{net} and B_{net} -percentile values that lie outside of the top 10: such structures are probably damaged, but are unlikely to display visible artefacts in their associated difference maps to enable "confirmation" of the B_{net} (-percentile) reading (hence the need to develop the metrics in the first place)."

REVIEWERS' COMMENTS

Reviewer #1 (Remarks to the Author):

The revisions has brought ample improvements to the manuscript and all point from the initial review were addressed. I recommend publication without additional changes changes.

Reviewer #2 (Remarks to the Author):

The authors have addressed most, but not all, of the previous review comments, so it is the opinion of this reviewer that the manuscript is still not ready for publication.

It is puzzling, in fact, that they seem to not really understand how to respond to reviewer comments. Lengthy paragraphs are written in the rebuttal to explain things to the reviewer alone. This goes as far as providing figures. And yet the manuscript itself was not updated in that case. The purpose of peer review is to identify problems with the document that other readers, perhaps 10 year from now, may find confusing, ambiguous or otherwise problematic. It is far more important to fix the manuscript than to treat the rebuttal as some kind of supplementary document. Add the figure to the Supp.

Another example is ambiguous language, such as the common mistake of writing "a number of" or, in this case, "numerous damaged structures" is always better written by replacing this text with the actual number that was determined in the work. Much to this reviewer's surprise, the authors simply refused to do this. There is always a statistically valid way to threshold a metric. In your analysis you encountered structures that your proposed metric said were "damaged". You should be able to count them. Report the number please. Even if it is "10". Not reporting this number casts serious doubts on the rigor of the reported work, not just in the eyes of this reviewer, but any future reader of the document.

The authors were also asked to use past tense to describe the work they are reporting. This is customary in scientific writing because using present or future tense is almost always uninformative to the reader. Scientific papers are not living documents, they are archival. Not using past tense makes it sound like the analysis is incomplete. Yes, it may be true the work is ongoing, but that is

always true. Tell the story you have, not the story that isn't finished. The relevant paragraph, even after being re-written, still reads like a proposal and not a report.

Another suggested correction was to avoid use of "It is widely accepted that", which invariably signals poor erudition. The suggestion was to cite a recent review, but instead they chose a 12-year-old paper. A more recent review, or perhaps a pair of papers, one very old and one very new, would much better support this assertion. And please remove "it is widely accepted that".

Overall, the authors chose to heed the majority of reviewer comments and suggestions, but dig in their heels on others that ought to be easy to fix and would have significantly strengthened the document. It is not entirely clear why, but it would appear that we have reached the limit of the impact advise from reviewers can have on this paper. Consequently, this reviewer has no further comments.

REVIEWERS' COMMENTS

Reviewer #1 (Remarks to the Author):

The revisions has brought ample improvements to the manuscript and all point from the initial review were addressed. I recommend publication without additional changes.

We thank the reviewer for the positive comments.

Reviewer #2 (Remarks to the Author):

The authors have addressed most, but not all, of the previous review comments, so it is the opinion of this reviewer that the manuscript is still not ready for publication.

It is puzzling, in fact, that they seem to not really understand how to respond to reviewer comments. Lengthy paragraphs are written in the rebuttal to explain things to the reviewer alone. This goes as far as providing figures. And yet the manuscript itself was not updated in that case. The purpose of peer review is to identify problems with the document that other readers, perhaps 10 year from now, may find confusing, ambiguous or otherwise problematic. It is far more important to fix the manuscript than to treat the rebuttal as some kind of supplementary document. Add the figure to the Supp.

We appreciate the point of view of the reviewer and regret that they are clearly irritated by our full responses and explanations. However, in our defence we would like to point out that the senior author has published over 180 peer reviewed papers, is one of three main Section Editors of a major crystallographic journal, and has herself supplied peer reviews on over 200 papers written by others. Thus we have some confidence that we usually understand how to respond to reviewers comments.

Another example is ambiguous language, such as the common mistake of writing "a number of" or, in this case, "numerous damaged structures" is always better written by replacing this text with the actual number that was determined in the work. Much to this reviewer's surprise, the authors simply refused to do this. There is always a statistically valid way to threshold a metric. In your analysis you encountered structures that your proposed metric said were "damaged". You should be able to count them. Report the number please. Even if it is "10". Not reporting this number casts serious doubts on the rigor of the reported work, not just in the eyes of this reviewer, but any future reader of the document.

Because we were reluctant to set a threshold for damage since it is a sliding scale rather than a step function with a clear value, we specifically do not want to put a number here. We have changed 'numerous damaged structures' to 'structures with a range of damage features' and changed 'not be identified as unusual' to 'remain unidentified' (to get the Abstract back to 150 words).

Note that we have now specified a threshold as originally requested by the second reviewer and we have added this on P 18, lines 390-392:

'Nevertheless, to aid the user, as a rough guide we suggest that a B_{net} value greater than 3.0, and/or a B_{net} -percentile value greater than 0.95, would merit further inspection of a structure.'

The authors were also asked to use past tense to describe the work they are reporting. This is

customary in scientific writing because using present or future tense is almost always uninformative to the reader. Scientific papers are not living documents, they are archival. Not using past tense makes it sound like the analysis is incomplete. Yes, it may be true the work is ongoing, but that is always true. Tell the story you have, not the story that isn't finished. The relevant paragraph, even after being re-written, still reads like a proposal and not a report.

We apologise that the reviewer found this frustrating. We have now changed P4 lines 103- 105 and added a Supplementary Figure 1 to show that it is necessary to use the aspartate and glutamate oxygens for the B_{net} calculation, rather than using all the atoms in the structure. Our new figure shows that the latter method would not adequately discriminate between low dose and high dose. The text has been changed from:

Accordingly, the sensitivity of a metric that measures the skewness of the B_{Damage} distribution of all atoms in a structure to specific radiation damage artefacts is suboptimal.

to:

Accordingly, the sensitivity of a metric that measures the skewness of the B_{Damage} distribution of all atoms in a structure to specific radiation damage artefacts was found to be suboptimal (Supplementary Figure 1).

Another suggested correction was to avoid use of "It is widely accepted that", which invariably signals poor erudition. The suggestion was to cite a recent review, but instead they chose a 12-year-old paper. A more recent review, or perhaps a pair of papers, one very old and one very new, would much better support this assertion. And please remove "it is widely accepted that".

Apologies, we have replaced the older review with the most recent one (Garman and Weik 2017) and removed the offending clause from P9, line 200.

Overall, the authors chose to heed the majority of reviewer comments and suggestions, but dig in their heels on others that ought to be easy to fix and would have significantly strengthened the document. It is not entirely clear why, but it would appear that we have reached the limit of the impact advise from reviewers can have on this paper. Consequently, this reviewer has no further comments.

We thank the reviewer for their time on our manuscript.